# The Effect of Full-Scale Exchange of Ca^2+^ with Zn^2+^ Ions on the Crystal Structure of Brushite and Its Phase Composition

**DOI:** 10.3390/biomimetics8040333

**Published:** 2023-07-28

**Authors:** Abdulaziz A. Alanazi, Fahad Abdulaziz, Mohammed Alyami, Satam Alotibi, Salah Sakka, Saida Abu Mallouh, Rund Abu-Zurayk, Mazen Alshaaer

**Affiliations:** 1Department of Chemistry, College of Science and Humanities in Al-Kharj, Prince Sattam bin Abdulaziz University, Al-Kharj 11942, Saudi Arabia; abdulaziz.alanazi@psau.edu.sa; 2Department of Chemistry, College of Science, University of Ha’il, Ha’il 81451, Saudi Arabia; fah.alanazi@uoh.edu.sa; 3Department of Physics, College of Science and Humanities in Al-Kharj, Prince Sattam bin Abdulaziz University, Al-Kharj 11942, Saudi Arabia; m.alyami@psau.edu.sa (M.A.); sf.alotibi@psau.edu.sa (S.A.); 4Department of Oral and Maxillofacial Surgery and Diagnostic Sciences, Prince Sattam bin Abdulaziz University, Al-Kharj 11942, Saudi Arabia; s.sakka@psau.edu.sa; 5Nanotechnology Center-Hamdi Mango Center for Scientific Research, The University of Jordan, Amman 11942, Jordan; saida387439@yahoo.com (S.A.M.); r.abuzurayk@ju.edu.jo (R.A.-Z.); 6Department Mechanics of Materials and Constructions (MEMC), Vrije Universiteit Brussels (VUB), Pleinlaan 2, 1050 Brussels, Belgium

**Keywords:** brushite, parascholzite, hopeite, biomaterials, SEM, Zn

## Abstract

This study was carried out to investigate the effect of a complete exchange of Ca^2+^ with Zn^2+^ ions on the structure of brushite (CaHPO_4_·2H_2_O), which might be advantageous in the production process of Ca_x_Zn_1−x_HPO_4_·nH_2_O. To acquire the starting solutions needed for the current study, (NH_4_)_2_HPO_4_, Ca(NO_3_)_2_·4H_2_O, and Zn(NO_3_)_2_·6H_2_O were utilized in several molar concentrations. The findings indicate that Ca is partly substituted by Zn when the Zn/Ca molar ratio is below 0.25 and that Zn doping hinders the crystallization of brushite. A continued increase in the Zn/Ca molar ratio to 1 (at which point the supersaturation of the Zn solution rises) led to a biphasic compound of monoclinic brushite and parascholzite precipitate. Elevating the Zn/Ca molar ratio to 1.5 resulted in a precipitate of a parascholzite-like mineral. Finally, increasing the Zn/Ca molar ratio to 4 and above resulted in the formation of the hopeite mineral. Future biomaterial production with specific and bespoke characteristics can be achieved by adjusting the Zn/Ca ratio in the starting solution. It Rhas been established that the Zn/Ca ratio in the starting solution can be adjusted to obtain minerals with specific compositions. Thus, new synthesis methods for parascholzite and hopeite were introduced for the first time in this manuscript.

## 1. Introduction

There is currently a great demand for the production of appropriate bone substitutes and implants for the treatment of various bone defects in vivo [1,2]. Bone defects are associated with several factors, the most important of which are osteoporosis, surgical operations, and fractures. Traditionally, bone tissue substitutes are based on the use of autografts as a standard for bone replacement [3]. However, there are several challenges when using these conventional bone implants and their substitutes such as the high risk of infection, graft material availability, and other complications [4]. Bone marrow treatment faces additional limitations such as limited donor sources and safety concerns [3]. To overcome these challenges, synthetic bone materials have been developed, which offer a viable alternative to autografts that can also reduce the risk of infection and other complications. There is also a growing need for clinically viable implants that are degradable in vivo. 

There are two common types of bone substitutes and implants: inert materials such as alloys and biodegradable materials such as (CaPs) calcium phosphates [3]. Inert or non-biodegradable materials have been used for decades in surgical operations to serve as bone substitutes or implants. The main disadvantage of using these materials is mainly the uncertainty of whether they can remain in the human body without causing infection or side effects, with the resulting problem that additional operations may be required to remove them after recovery. Biodegradable materials are the most promising bone substitutes because they are more suitable for different clinical applications such as bone fracture treatment [1,3]. These materials can be resorbed by the human body and are then replaced with natural bone tissue over time, meaning that additional surgery can be avoided. The main distinguishing features of these biodegradable materials are their osteogenic, osteoinduction, and osteoconduction capabilities. However, a major issue with using biodegradable materials for bone regeneration is their ability to integrate with and become vascularized in the body [1].

Calcium phosphate is an important material in the fields of medicine, biology, engineering, and environmental science, and acts as a crucial component in many applications. Its functionality depends on its structural characteristics [1,2,3,4]. CaPs offer advantageous mineralogical and biochemical properties. They offer particularly good prospects for use in medical applications because they are biocompatible, less toxic, highly bioactive, and resemble the mineral phases naturally found in osseous tissue [4,5,6]. Currently, CaPs can be used to manufacture sophisticated biocements and bioceramics for use in dentistry and medical interventions [3,7,8]. Previous research suggests that CaPs can be used as fertilizer [9] as well as in bone tissue engineering [7,10,11], medication delivery [12], and construction [13,14,15].

In recent decades, one of the most significant CaPs applications identified by research has been its use in the enhancement of artificial implant design to accelerate and improve bone healing [16,17]. Many biological assessments and applications of CaPs-based coatings have been carried out in vitro. The surface modification and engineering of an implant is an essential method of improving the osteoconductive and biocompatible characteristics of different medical devices; this is because the implant surface is the first part of the implant that interacts with the host. Coating an artificial implant with a layer of CaPs has also been confirmed as an effective strategy to provide a biomaterial with enhanced biocompatibility and acceptable osteoconductivity [18]. Many physicochemical techniques have been employed in the deposition of CaPs coatings including electrophoretic deposition, plasma spraying, and immersion. Plasma spraying is an effective and low-cost technique that is widely used to deposit CaPs onto orthopedic implants [16]. 

Homogeneous and heterogeneous CaPs are formed in supersaturated solutions. Several phases may develop from a calcium phosphate solution that is supersaturated. These occur in the following order of reduced solubility: amorphous calcium phosphate (ACP) or hydroxyapatite (Ca_10_(PO_4_)_6_·(OH)_2_, HA), brushite (CaHPO_4_·2H_2_O, DCPD), monetite (CaHPO_4_, DCPA) [13,16], β-tricalcium phosphate (Ca_3_(PO_4_)_2_, (β-TCP)) [19,20,21], and octacalcium phosphate (Ca_8_H_2_(PO_4_)_6_·5H_2_O, OCP) [8,21,22]. The solubilities and crystalline structures of the individual phases differ noticeably from one another. In aqueous supersaturated solutions, the calcium phosphate phases may either form spontaneously or with the assistance of a substrate. This method is known as seeded crystal growth and occurs when the substrate contains the calcium phosphate phase that will later grow.

According to several research studies, one key material is dicalcium phosphate dehydrate (DCPD), also known as brushite, which has the formula CaHPO_4_·2H_2_O [8]. This element serves as a precursor for bioceramics, bone cement, and hydroxyapatite (HA). The brushite that is found in mineralized tissues has numerous potential applications in the medical field, particularly in the manufacture of bone cement [23,24,25,26,27,28]. It has been proven to be stable in pH conditions ranging from 4.0 to a6.5 and at low temperatures [29]. Additionally, it can be reabsorbed promptly by the human body, supporting bony tissue remodeling, since it is typically metastable at a pH of around 7.4 [30].

Zinc is a heavy metal that is typically found in nature in its divalent state. It is regarded as an essential mineral that is necessary for the body’s manufacturing of hundreds of enzymes [23]. The normal concentration of zinc in serum is between 109 and 130 micrograms per deciliter; daily recommended intakes of zinc vary according to the patient population [24]. Moreover, zinc is a cofactor in numerous enzymatic processes involving DNA expression, membrane stability, vitamin A metabolism, and the gustatory and olfactory systems [25,26], and it plays a significant role in fetal development. Another characteristic of zinc’s importance in the human body is its inverse relationship to copper levels [27,28,29]. Zn is considered as one of the most important trace elements and contributes to bone growth and differentiation. The average content of Zn in the human body is 0.0172 wt%, and around 28% of this Zn content is present in the tissues [30]. Zinc deficiency slows bone growth and adversely affects bone metabolism. Zinc deficiency, on the other hand, is a risk factor for osteoporosis. It is also an important trace element for promoting functional bone cell (osteoblast) differentiation, growth, and proliferation. Finally, zinc is one of the components of alkaline phosphatase and other proteins and metalloenzymes. 

One of the main functions of a bone substitute is to mimic the growth of natural bone tissue, thus, growth factors are usually added to bone cement. However, these growth factors are usually associated with high costs and can cause unwanted side effects. As an alternative to growth factors, ions such as Zn^2+^ can be added to bone cement; this alternative is characterized by lower costs and has less potential for side effects in the human body. As shown above, Zn plays a major role in the human body, therefore, introducing Zn^2+^ in this context should have an additional positive effect and stimulate natural bone formation. In addition, Zn is known to possess antibacterial properties [30]. Therefore, adding Zn to bone cement may yield antimicrobial characteristics, and thus avoid infections during and after surgery.

Several studies have considered the possibility of substituting Ca with Zn in different calcium phosphate minerals [23,24,25,26,27]. The different Zn-substituted CaP formulations were prepared and characterized. It was found that the substitution of Ca by Zn up to 10 atomic % in HA did not affect the HA and brushite crystal structures [23]. The Zn ions can replace the Ca ions in the β-TCP structure up to 10 atomic %, with the preference seen in the Ca(5) site. Zn-substituted brushite cements exhibit an observable performance in terms of bone restoration and antibacterial activity. Elsewhere, it was observed that only 0.6 wt% of Zn substitution led to an inhibitory effect regarding some types of bacteria such as *Escherichia coli.* It was also found that the optimum substitution of Ca with Zn in α-Zn-TCP was around 0.03 wt% [29]. The resulting calcium phosphate material was able to stimulate additional bone formation compared to the pure α-TCP.

Previous investigations have explored how ionic replacement affects the morphological structure and hydrolysis of brushite [31]. Accordingly, the spectrum of potential replacements would vary, depending on the doped ions [32]. Although initial research has demonstrated that Zn replacement could be integrated with the solid phase of brushite, a mixture of phases has been found with higher Zn concentrations in the original solution [27].

The research team was motivated to investigate the crystal shape, the physical and chemical characteristics, and the mineral properties of the compounds that form when Zn ions gradually replace the Ca ions (from 0% to 100%) in brushite. The results could help fill the research gap regarding mineral synthesis in the production of biomaterials that are intended for a range of uses in the pharmaceutical and bone tissue engineering industries.

## 2. Materials and Methods

### 2.1. Materials 

Diammonium hydrogen phosphate (NH_4_)_2_HPO_4_ was obtained from Techno Pharmchem, Delhi, India, while calcium nitrate tetrahydrate (Ca(NO_3_)_2_·4H_2_O) and zinc(II) nitrate hexahydrate (Zn(NO_3_)_2_·6H_2_O) were acquired from LOBA Chemie, Mumbai, India. Distilled water (0.055 µS/cm) was prepared using a purification system (PURELAB option-Q, ELGA, UK). A digital analytical balance (EX324N, OHAUS, Parsippany, NJ, USA) and a magnetic stirrer (ISOTEMP, Fisher Scientific, Shanghai, China) were employed as needed. 

### 2.2. Synthesis of Ca_x_Zn_1−x_HPO_4_·nH_2_O Compounds

Seven Ca_x_Zn_1−x_HPO_4_·nH_2_O compounds were made at room temperature (RT), which required the use of (NH_4_)_2_HPO_4_, Ca(NO_3_)_2_·4H_2_O, and Zn(NO_3_)_2_·6H_2_O 0.5 moL/L solutions in the molar proportions shown in Table 1.

To create unpolluted brushite (shown in Table 1 as BZn0), 100 mL of a Ca(NO_3_)_2_·4H_2_O solution was added at an ~2 mL/min flow rate to the (NH_4_)_2_HPO_4_ solution using a glass funnel with a stopcock while stirring at 450 rpm until the 1 Ca/P molar ratio was reached within one hour. To ensure a fully homogenized solution, it was stirred at RT for 60 min. The pH was fixed to 6−6.5 by adding ammonia (~15 moL/L, Labochemie, Mumbai, India). The required white precipitate was attained after the solution was vacuum-filtered using a Büchner funnel and qualitative filter paper (45 µm, Ø 12 cm, Double Rings, China). To avoid agglomeration, the filter cake was cleansed, first in deionized water and then in ethanol; the cleansing was performed three times for each powder [33,34,35,36]. Finally, the sample was dried in an oven (40 °C) for 7 days after being placed on a watch glass (ED53/E2, Binder, Tuttlingen, Germany) [37]. 

To obtain the BZn2, BZn4, BZn5, BZn6, and BZn10 compounds, the Ca(NO_3_)_2_·4H_2_O and Zn(NO_3_)_2_·6H_2_O solutions were mixed in the molar ratios listed in Table 1. As described above, 100 mL of the resulting solution was added at a flow rate of ~2 mL/min to 100 mL of the (NH_4_)_2_HPO_4_ solution. To create BZn10, the same procedure was completed after mixing (NH_4_)_2_HPO_4_ and Zn(NO_3_)_2_·6H_2_O. 

The schematic diagram of the experimental procedure and the preparation of the compounds/minerals is illustrated in Figure 1.

### 2.3. Characterization Techniques

Qualitative mineralogical analysis was performed on the BZn0−BZn10 samples using a Shimadzu XRD diffractometer-6000 (Kyoto, Japan) with a cobalt tube and a 2-theta scanning range of 10−60° at a 2°/min scan rate. Match! software (version 3.15, Crystal Impact, Bonn, Germany) was used for phase and crystal analysis using powder XRD diffraction data. Complete product morphology entailed performing scanning electron microscopy with the Inspect F50 (The Netherlands); an XPS system (Thermo K Alpha spectrometer, Waltham, MA, USA) was employed to conduct X-ray photoelectron spectroscopy in order to control the surface chemistry and complete the elemental sample analysis. Finally, a thermogravimetric (TG) analyzer (Netzsch, Waldkraiburg, Germany, TG 209 F1 Libra) was employed to determine the mass loss (~100 mg) resulting from heating each product from 40 °C to 750 °C in 5 °C/min^−1^ increments under a helium atmosphere. Inductively coupled plasma optical emission spectrometry (ICP-OES) (Thermo Fisher Scientific, Waltham, MA, USA, iCAP 7000 series) was used to determine the Zn contents of the synthesized compounds. 

## 3. Results and Discussion 

### 3.1. Mineralogical and Microstructural Analysis

Figure 2 shows the XRD scan analyses of all of the samples as well as that of pure brushite (CaHPO_4_·2H_2_O) or BZn0. The qualitative mineralogical analysis findings confirmed that pure brushite (BZn0) was obtained by mixing the solutions of Ca(NO_3_)_2_·4H_2_O and (NH_4_)_2_HPO_4_ with Ca to P molar ratio of 1:1, as shown in Table 1. After nucleation, the crystals expanded proportionally in the three main crystallographic planes with Miller indices of (020), (121-), and (141-). Additionally, all of the peaks identified in the pattern as being made by BZn0 suggest the monoclinic structure of brushite [21,37]. The XRD peak at a 2-theta of 11.7° chiefly implies advanced crystal growth alongside the (020) plane [7]. 

The XRD pattern obtained for BZn2 had an amorphous structure. Here, the crystallinity loss caused by the initiated Zn^2+^ ions might be linked to the smaller Zn^2+^ ionic radius (74 p.m.) compared to Ca^2+^ (99 p.m.), which was, in part, substituted by the external ions [25]. Zn^2+^ ions are often involved in the different mechanisms comprising the initiation of Zn^2+^ into the crystal structure of CaPs [38]. 

The above findings suggest that Zn was partially initiated into the brushite structure and visibly impacted its lattice. Therefore, it can be considered to be an inhibitor of the crystalline phase of calcium phosphate formation [39]. The monoclinic crystal structure of brushite with a Zn substitution below the molar ratio of 0.25 was evidenced by the lattice parameters and the total size of the brushite’s unit cell, Table 2. 

As indicated by the patterns produced by BZn4 (which are related to the Zn/Ca molar ratio of 0.67), a new phase of CaZn_2_(PO_4_)_2_·2H_2_O formed: parascholzite. The corresponding planes were (200), (110), (31-1), (220), and (420) [40]. The BZn series, with an equal molar ratio of Zn/Ca, demonstrated the formation of a biphasic compound comprising brushite (21%) and parascholzite (79%), Table 2. The brushite disappeared again at a Zn/Ca molar ratio of 1.5, BZn6, and only CaZn_2_(PO_4_)_2_·(H_2_O)_2_ precipitated. Finally, a monophasic phase of hopeite (H), Zn_3_(PO_4_)_2_·4H_2_O, was formed by increasing the Zn/Ca ratio above 1.5, producing BZn8 and BZn10.

Figure 3 shows the SEM images of brushite (BZn0) as well as those of the Ca_x_Zn_1−x_HPO_4_·nH_2_O compounds (BZn2, BZn4, BZn5, BZn6, BZn8, and BZn10), based on various Zn/Ca molar ratios. As shown by Point 1 in Figure 3A (BZn0, with an equal Ca to P molar ratio), the precipitation of monophase brushite eventually occurs with monoclinic crystals, as is to be expected at the pH utilized in this work [8,19]. The monoclinic crystals had dimensions of 0.5 × 5 × 10 µm^3^, similar to those acquired and reported by other investigators [15]. Furthermore, flat-monoclinic platy morphology is a conventional crystal structure for precipitated brushite [41]. It is clear from Point 2 in Figure 3B that since the Ca substitution by Zn occurred at a level up to 20%, forming BZn2, the formation of an amorphous structure is in good agreement with the XRD patterns seen in Figure 2. Monoclinic parascholzite crystals could be observed in Figure 3C, formed by increasing the Zn/Ca molar ratio to 0.67 for BZn4. Fine biphasic monoclinic crystals of brushite and parascholzite were precipitated with a 50% Ca substitution by Zn, creating BZn5 (Figure 3D). A spherical agglomeration of nano-sized monoclinic CaZn_2_(PO_4_)_2_·(H_2_O)_2_ crystals could be observed when increasing the Zn/Ca molar ratio to 1.6, forming BZn6 (Figure 3E). In turn, Figure 3F shows BZn8, demonstrating the precipitation of nanosized orthorhombic hopeite crystals, the parameters of which are indicated in the XRD pattern obtained for BZn2, which has an amorphous structure. Here, the crystallinity loss caused by the initiated Zn^2+^ ions might be linked with the smaller Zn^2+^ ionic radius (74 p.m.) compared with that of Ca^2+^ (99 p.m.), which was, in part, substituted by a foreign dopant [25]. Zn^2+^ ions are often involved in the different mechanisms of the initiation of Zn^2+^ into the crystal structure of CaP [38]. 

The above findings suggest that Zn was partially initiated into the brushite structure and visibly impacted its lattice. Thus, Zn can be considered to be an inhibitor of the crystalline phase of calcium phosphate formation [39]. The monoclinic crystal structure of brushite, with Zn substitution below the molar ratio of 0.25, is reported in [25]. The lattice parameters and the total size of the brushite’s unit cell were similar to that of standard brushite [40].

As a result of increasing the Zn content from 80% for BZn8 to 100% for BZn10, large orthorhombic crystals of hopeite formed (with dimensions of around 20 µm × 30 µm × 2 µm), as depicted in Figure 3F,G. This proves that the entire substitution of Ca by Zn resulted in a monophasic compound containing only orthorhombic HZnP nano- (BZn8) and microcrystals (BZn10). Therefore, the results generated by the SEM analysis validate those that were acquired via XRD, as shown in Figure 2. 

The elemental analysis shows that the Zn content presented an incremental trend, as reported in Figure 4. In general, there was a clear link between the gradual increase in Zn seen in the resulting compounds, corresponding to the proportions of Zn in the initial solutions. These results confirm that Zn was an essential part of the resulting compounds and had a role in the reactions at various stages. 

### 3.2. Chemical Composition and Elemental Analysis of Ca_x_Zn_1−x_HPO_4_·nH_2_O Powders

The prepared Ca_x_Zn_1−x_HPO_4_·nH_2_O compounds were analyzed via XPS to assess the influence of the Zn/Ca ratio that existed in the initial solution on the chemical states of Zn, Ca, and P; the surface chemistry regarding the chief elements of the synthesized Ca_x_Zn_1−x_HPO_4_·nH_2_O compounds and the related results are shown in Figure 5 [40,41,42]. Again, the Zn/Ca molar ratio was found to impact the degree of replacement of Ca with Zn seen in the brushite, in addition to the precipitation of the compounds (with elevated ratios of Zn/Ca). The peaks relating to the Zn 2p3 and Zn 2p1 orbital were clear in the XPS spectra of compounds BZn2 to BZn10. The concentrations gradually increased up until BZn5, while the intensity of the Ca 2p and Ca 2s peaks declined in a manner closely corresponding to the rise in the Zn/Ca ratio. Conversely, the intensity of the P 2s peak remained almost unchanged, indicating that the concentrations of the P, Ca, and Zn peaks relied on the degree of Ca substitution with Zn, in addition to the amounts of the precipitated compounds (with greater Zn/Ca molar ratios).

The impact of the Zn/Ca ratio on the binding energies of Zn 2p, Ca 2s, and P 2s and the recorded peaks can be seen in Figure 6A–C, respectively. These results demonstrate that in the case of the Zn/Ca 0.25 (BZn2) molar ratio, the binding energies of the P 2s and Ca 2s peaks rose from 439 eV to 443 eV and from 190 eV to 195 eV, respectively [40]. Meanwhile, the peaks relating to Ca 2s and Ca 2p disappeared from the BZn8 compound. The binding energies corresponding to Zn 2p3 and Zn 2P1 were observed at 1026 eV and 1048 eV for those compounds with Zn contents (BZn2 to BZn10) [41]. 

The related assessments offer further evidence of how the Zn/Ca ratio rose to 1 (BZn5) and then decreased (BZn6–BZn10). The peaks corresponding to Ca 2s that appeared in BZn4–BZn6 indicate the precipitation of CaZn_2_(PO_4_)_2_·2H_2_O. However, these peaks disappeared with higher Zn/Ca molar ratios (BZn8 and BZn10) because of the formation of Zn_3_(PO_4_)_2_·4H_2_O.

Altogether, the XPS findings verify that when the Zn/Ca molar ratio of the starting solution is minimal (BZn2–BZn6), Ca is partly substituted by Zn. However, to modify the crystal structure of the Ca_x_Zn_1−x_HPO_4_·nH_2_O compounds, it suffices to raise the binding energies of the Ca 2s and P 2s peaks. As the Zn intensity rises with the reduction in Ca concentration, the level of supersaturation decreases (increases) with respect to brushite. Consequently, a pure Zn_3_(PO_4_)_2_·4H_2_O compound is obtained when low Ca (under 20%) is available, as is the case with the BZn8 and BZn10 compounds [42]. 

The XPS findings confirm that the presence of Zn was identified in compounds BZn2 to BZn10. Although the Zn/Ca molar ratio was set at 4 in the starting solutions, no Ca ions were detected in BZn8, which is in good agreement with the XRD analysis shown in Figure 2 [43]. This offers a strong indication that the Ca ions are not involved in the synthesis of hopeite or any other compound if the Zn/Ca molar ratio is greater than 1.5. Finally, the shifts in the peaks of Ca and P to higher positions as a result of adding Zn in the starting solutions (BZn2–BZn10) is an indication of changes to the areas surrounding these elements, caused by transforming the brushite into new minerals (parascholzite or hopeite).

### 3.3. Thermogravimetric Analysis (TGA) 

The TGA analysis outcomes for the investigated materials (BZn0 to BZn10) are depicted in Figure 7. The crystal structure of the brushite comprises compact analogous chains, in which the Ca ions are organized into groups of six PO_4_ ions and two O_2_ atoms that belong to the structural water in brushite [14]. Furthermore, as confirmed by two sharp peaks corresponding to the mass loss caused by increasing the temperature from 80 °C to 220 °C, brushite was identified by the two structural water molecules found in its lattice and the adsorbed water molecules on its surface [36,44]. The existing evidence suggests that part of the chemically attached water is released when the brushite converts to monetite (CaHPO_4_) at ~220 °C [31] and converts further to calcium pyrophosphate (Ca_2_P_2_O_7_) as the temperature increases to ~400 °C [13]. According to the results obtained in the present study, when pure brushite (BZn0) is heated to 750 °C, roughly 21 wt% of its mass is lost [15], which is similar to the hypothetical mass loss of 20.93 wt% [45]. In contrast, the BZn2−BZn10 samples, with their increasingly higher Zn/Ca ratios, lost a lower mass (16−10%).

The dehydration reaction of brushite is given in Equation (1), while the formation of calcium pyrophosphate can be explained using Equation (2).
(1)$${CaHPO}_{4}\cdot 2H_{2}O\to {CaHPO}_{4}+2H_{2}O$$
(2)$${2CaHPO}_{4}\to {Ca}_{2}P_{2}O_{7}+H_{2}O$$

Figure 8 shows the mass loss rate for the Ca_x_Zn_1−x_HPO_4_·nH_2_O compounds as a function of temperature. In an earlier study [36], as shown in Figure 8A−G, the dehydration peaks related to the two structural water molecules of pure brushite (BZn0) were clearly visible. When the Ca was substituted by 20% of Zn in the starting solution, BZn2 was formed (see Figure 8B). However, one main zone of mass loss was attained at roughly 93 °C. The absence of mass loss peaks at higher temperatures is indicative of the amorphous structure, as evident by the XRD patterns in Figure 8. All of the other samples—BZn4, BZn5, BZn6, BZn8, and BZn10—exhibited at least two major mass loss peaks. 

### 3.4. Phase Evolution during the Precipitation of Ca_x_Zn_1−x_HPO_4_·nH_2_O Compounds

The results of the current study highlight the finding that the unique profile of plate-like brushite crystals was conserved when the Zn/Ca ratio in the initial solutions did not surpass 0.25 [46]. Conversely, their dimensions declined because of the presence of Zn. Zinc impedes the crystal growth of brushite [47]; hence, eventually, the brushite crystals disappeared when the Zn/Ca ratio rose to 0.25 (BZn2), giving rise to an amorphous structure. The parascholzite mineral, CaZn_2_(PO_4_)_2_·2H_2_O, exhibited monoclinic crystals that precipitated as the only phase when the Zn/Ca ratio increased even more to 0.67 (BZn4). Biphasic powder results from both brushite and parascholzite, with a Zn/Ca molar ratio of 1 (BZn5). CaZn_2_(PO_4_)_2_·(H_2_O)_2_ was formed by raising the Zn/Ca molar ratio to 1.5. Finally, nano (BZn8) and micro (BZn10) orthorhombic hopeite (Zn_3_(PO_4_)_2_·4H_2_O) phases occurred with the presence of Zn/Ca molar ratios greater than 4 (Table 3). 

## 4. Conclusions

The substitution of Ca with Zn in brushite was studied by examining a range of Ca_x_Zn_1−x_HPO_4_·nH_2_O biomaterials, which underwent XRD, SEM, XPS, ICP, and TG analyses. The findings indicate that when the Zn/Ca molar ratio was below 0.25 in the starting solution, Ca was partly substituted by Zn. Increasing the Zn/Ca molar ratio to 0.2 hindered the crystallization of brushite; instead, a compound with an amorphous structure was precipitated. The parascholzite mineral, which exhibited a monoclinic crystal structure, precipitated at a Zn/Ca molar ratio of 0.67. When the Zn/Ca molar ratio progressively increased to 1 (whereby the solution’s supersaturation level rose with respect to Zn), a biphasic compound of monoclinic brushite and parascholzite precipitated. Increasing the molar ratio of zinc to calcium to a level of 1.5 resulted in the precipitation of a parascholzite-like mineral. Finally, raising the Zn/Ca molar ratio to a level of 4 and above led to the formation of the mineral hopeite, which has an orthorhombic crystal structure. The fine crystals (~0.5 µm) precipitated first, then increased in size as the Zn/Ca molar ratio increased. These findings may be beneficial for the forthcoming production of biomaterials with specific structural and bespoke properties since they suggest that by controlling the Zn/Ca ratio in the starting solution, a selection of material components and geomorphologies can be achieved. However, further research is needed to evaluate the functioning of the generated biomaterials, particularly their biological and antibacterial properties. Moreover, the influence of doping with added ions should be thoroughly examined since Zn may demonstrate either useful effects or harmful impacts on biomaterials that have been designed for medical purposes.

## Figures and Tables

**Figure 1 biomimetics-08-00333-f001:**
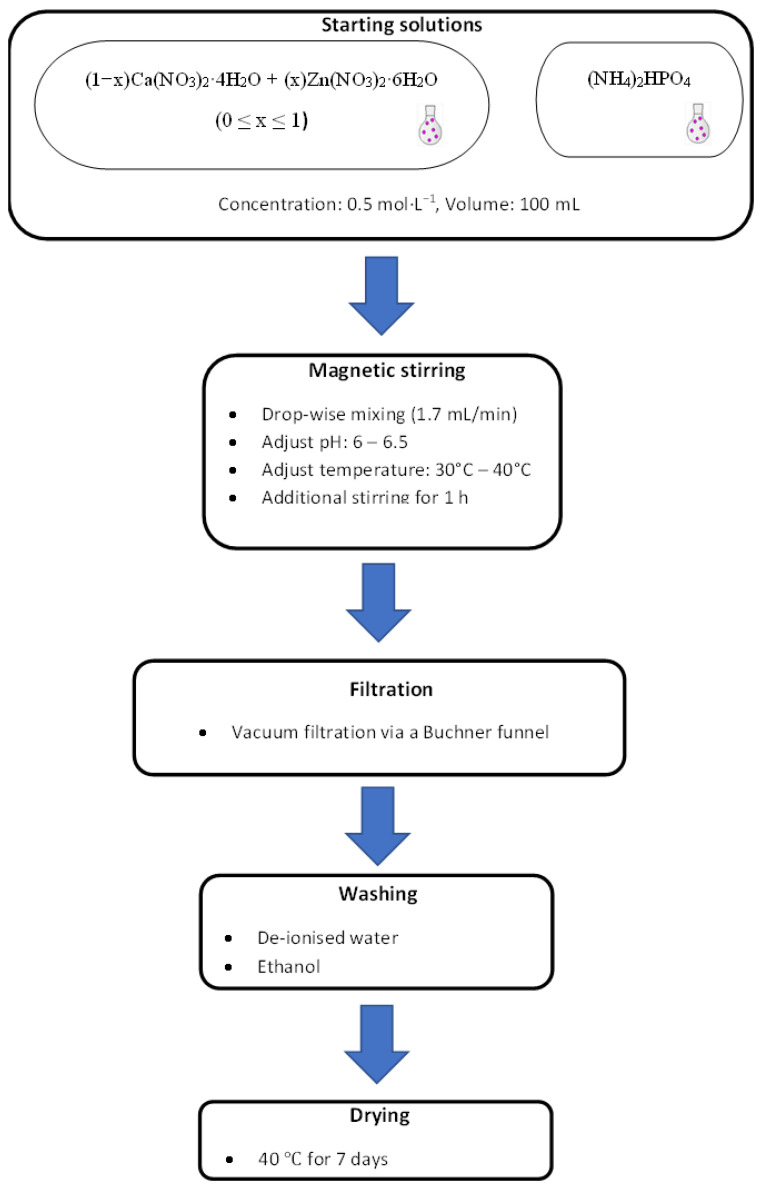
Experimental procedure for the synthesis of the compounds.

**Figure 2 biomimetics-08-00333-f002:**
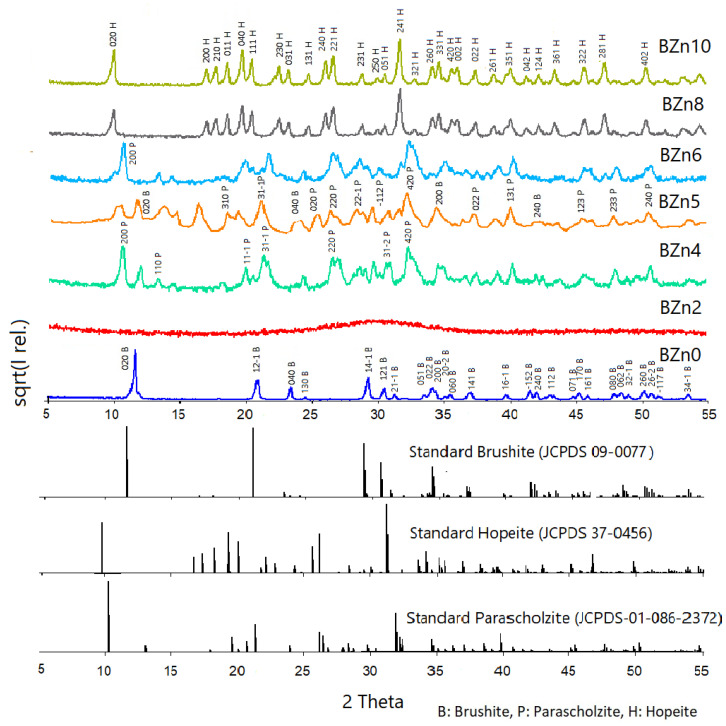
X-ray diffraction patterns of the Ca_x_Zn_1−x_HPO_4_·nH_2_O compounds produced under the conditions shown in Table 1.

**Figure 3 biomimetics-08-00333-f003:**
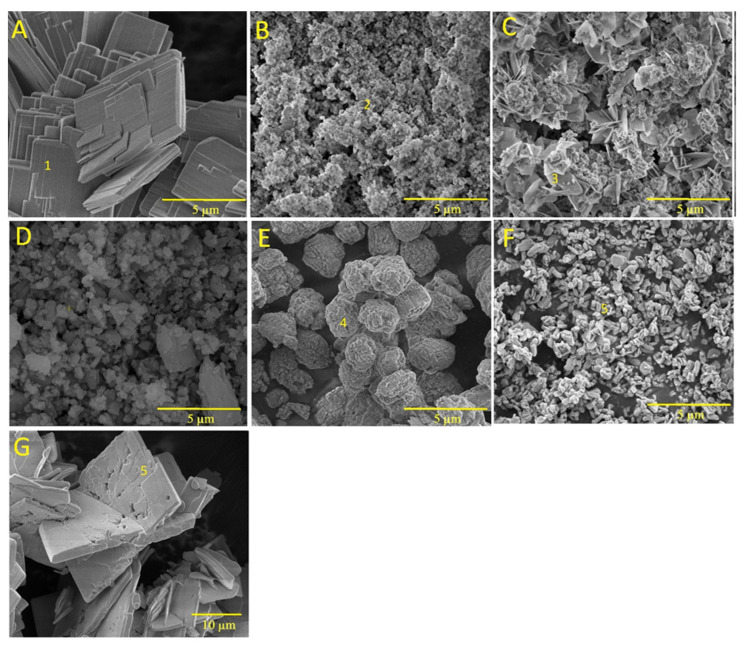
SEM images of the Ca_x_Zn_1−x_HPO_4_·nH_2_O compounds (labeled using the compound names shown in Table 1). (**A**) BZn0; (**B**) BZn2; (**C**) BZn4; (**D**) BZn5; (**E**) BZn6; (**F**) BZn8; (**G**) BZn10. Point 1, brushite crystals; point 2, amorphous structure; point 3, parascholzite crystals; point 4, CaZn_2_(PO_4_)_2_·(H_2_O)_2_ crystals; point 5, hopeite crystals.

**Figure 4 biomimetics-08-00333-f004:**
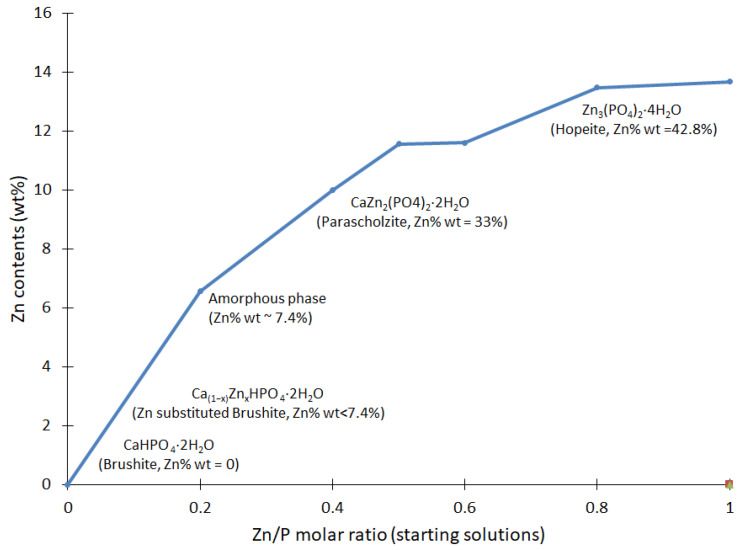
Analysis of the Zn contents in the Ca_x_Zn_1−x_HPO_4_·nH_2_O compounds, established via ICP-OES. The Zn/Ca molar ratio of the starting solutions is reported according to the values given in Table 1. The resulting mineral is shown as a function of the zinc contents.

**Figure 5 biomimetics-08-00333-f005:**
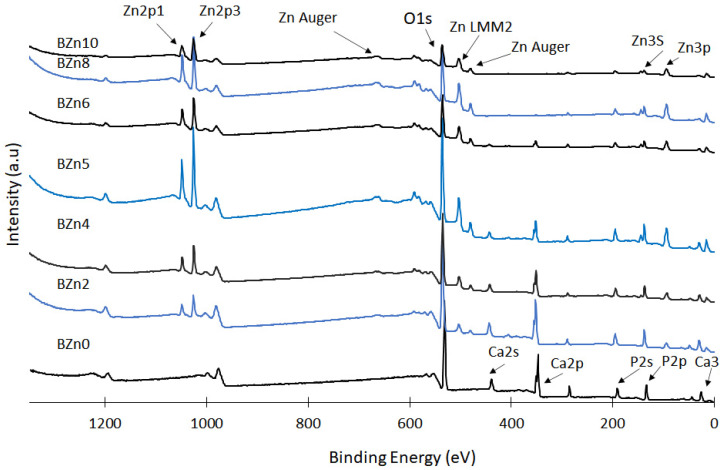
XPS spectra of the Ca_x_Zn_1−x_HPO_4_·nH_2_O compounds.

**Figure 6 biomimetics-08-00333-f006:**
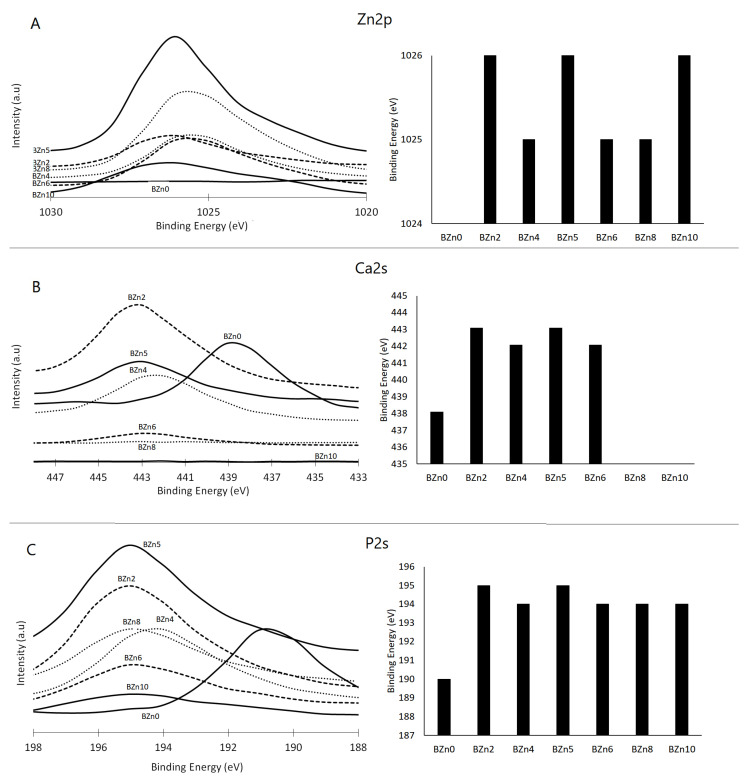
XPS analysis of the chemical state of (**A**) Ca 2s, (**B**) P 2s, and (**C**) Zn 2p orbitals in the Ca_x_Zn_1−x_HPO_4_·nH_2_O compounds.

**Figure 7 biomimetics-08-00333-f007:**
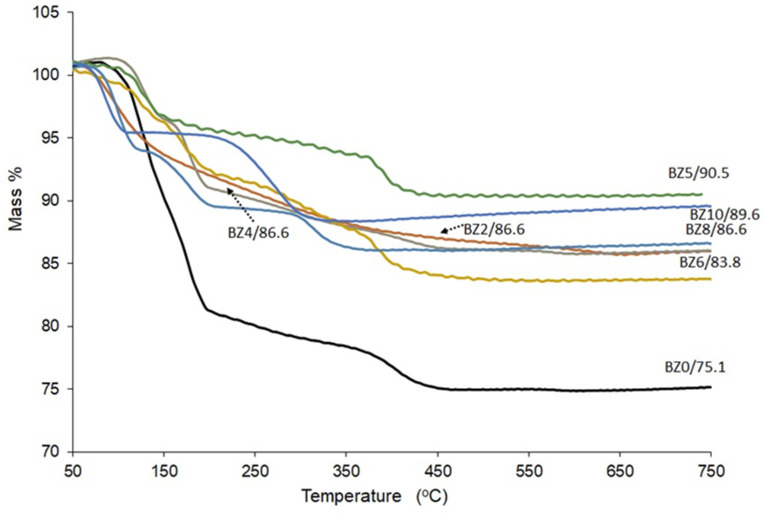
TG curves of the Ca_x_Zn_1−x_HPO_4_·nH_2_O compounds (product names BZn0–BZn10).

**Figure 8 biomimetics-08-00333-f008:**
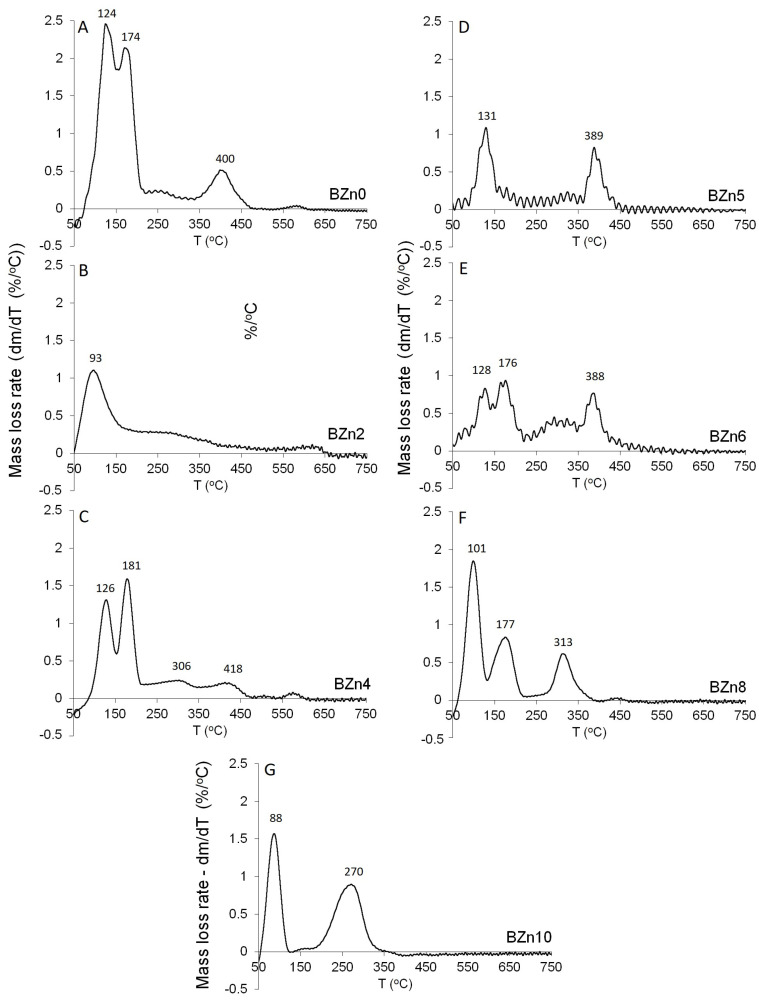
Differential TGA of the different Ca_x_Zn_1−x_HPO_4_·nH_2_O compounds; (**A**) BZn0, (**B**) bzN2, (**C**) BZn4, (**D**) BZn5, (**E**) BZn6, (**F**) BZn8, and (**G**) BZn10.

**Table 1 biomimetics-08-00333-t001:** Molar proportions of (NH_4_)_2_HPO_4_, Ca(NO_3_)_2_·4H_2_O, and Zn(NO_3_)_2_·6H_2_O, in addition to the Zn/Ca molar ratios applied for the synthesis of Ca_x_Zn_1−x_HPO_4_·nH_2_O compounds.

ID	(NH_4_)_2_HPO_4_	Ca(NO_3_)_2_·4H_2_O	Zn(NO_3_)_2_·6H_2_O	Zn/Ca Molar Ratio
BZn0	1	1	0	0
BZn2	1	0.8	0.2	0.25
BZn4	1	0.6	0.4	0.67
BZn5	1	0.5	0.5	1.0
BZn6	1	0.4	0.6	1.5
BZn8	1	0.2	0.8	4
BZn10	1	0	1	-

**Table 2 biomimetics-08-00333-t002:** The parameters of the brushite unit cell as determined by the XRD scans (Rietveld refinement, MATCH! Software analysis). * Not available.

ID	Crystal System	Phase Composition	Chemical Formula	Space Group	wt%
BZn0	Monoclinic	Brushite (B)	CaHPO_4_·2H_2_O	I 1 a 1	100
BZn2	Amorphous	*	*	*	100
BZn4	Monoclinic	Parascholzite (P)	CaZn_2_(PO4)_2_·2H_2_O	C 1 2/c 1	100
BZn5 (P)	Monoclinic	Parascholzite (P)	CaZn_2_(PO4)_2_·2H_2_O	C 1 2/c 1	79
BZn5 (B)	Monoclinic	Brushite (B)	Ca_x_Zn_x−1_HPO_4_·2H_2_O	I 1 a 1	21
BZn6	Monoclinic	*	Ca.Zn_2_(PO_4_)_2_ (H_2_O)_2_	I 1 2/c 1	100
BZn8	Orthorhombic	Hopeite (H)	Zn3(PO_4_)_2_·4H_2_O	P n m a	100
BZn10	Orthorhombic	Hopeite (H)	Zn_3_(PO_4_)_2_·4H_2_O	P n m a	100
	The lattice parameters				
ID	a (Å)	b (Å)	c (Å)	ß°	V (Å^3^)
BZn0	5.8151	15.2179	6.2664	116.413	496.649
BZn2	*	*	*	*	*
BZn4	17.863	7.412	6.667	106.25	847.4499
BZn5 (P)	17.863	7.412	6.667	106.25	847.4499
BZn5 (B)	5.8091	15.1656	6.2259	116.407	491.2613
BZn6	17.186	7.413	6.663	95.39	845.1116
BZn8	10.629	18.339	5.04	*	982.4232
BZn10	10.629	18.339	5.04	*	982.4232

* Unavailable or N/A.

**Table 3 biomimetics-08-00333-t003:** The crystal size and structure, along with the phase composition, as a function of the Zn/Ca molar ratio in the starting solutions.

Zn/Ca Ratio	0	0.25	0.67	1	1.5	4	Ca = 0
Phase/s	Brushite	-	Parascholzite	Brushite + Parascholzite	Parascholzite-like mineral	Hopeite	Hopeite
Crystal structure	Mono.	-	Mono.	Mono. + Mono.	Mono.	Ortho.	Ortho.
Crystal size (µm)	~10	-	~2	~0.5 + ~0.1	~2	~0.5	~30

## Data Availability

The data that support the findings of this study are available from the corresponding author, (Mazen Alshaaer), upon reasonable request.

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
