# Peer review of "The Effect of Full-Scale Exchange of Ca2+ with Zn2+ Ions on the Crystal Structure of Brushite and Its Phase Composition"

_biomimetics, 2023, doi:10.3390/biomimetics8040333_

Round 1

Reviewer 1 Report

The manuscript presents the effect of Zn/Ca ratios on crystallization of its phosphates at pH range of 6-6.5. The phases, morphologies, surface chemistry bonding are discussed. Considering many mistakes at present state, I think it need a major revision before it can be published by the journal.

---throughout the context, the source of phosphate, NaH2PO42H20 was wrongly treated as (NH4)2PO4.

---there are many mistakes in the context, e.g., superstation in line 32, different Z in line 124, and etc.

---the guiding descriptions in the template should be avoided, lines from 153-167.

---in the XRD patterns for BZn 8 and BZn10, one can observe many unassigned reflections around 35°, 42°, inconsistent to the assertion of phase-pure hopeite mentioned.

---for review in lines from 123-132, relevant Refs are needed.

The manuscript presents the effect of Zn/Ca ratios on crystallization of its phosphates at pH range of 6-6.5. The phases, morphologies, surface chemistry bonding are discussed. Considering many mistakes at present state, I think it need a major revision before it can be published by the journal.

---throughout the context, the source of phosphate, NaH2PO42H20 was wrongly treated as (NH4)2PO4.

---there are many mistakes in the context, e.g., superstation in line 32, different Z in line 124, and etc.

---the guiding descriptions in the template should be avoided, lines from 153-167.

---in the XRD patterns for BZn 8 and BZn10, one can observe many unassigned reflections around 35°, 42°, inconsistent to the assertion of phase-pure hopeite mentioned.

---for review in lines from 123-132, relevant Refs are needed.

Author Response

We thank the reviewer for her/his careful reading of the manuscript and her/his constructive remarks. We have taken the comments on board to improve and clarify the manuscript. Please find below a detailed point-by-point response to all comments.

  1. Throughout the context, the source of phosphate, NaH2PO42H20 was wrongly treated as (NH4)2PO4.
  • We apologize for this error. (NH4)2HPO4 was used as the P ion source. So, the text has been corrected accordingly.
  1. there are many mistakes in the context, e.g., superstation in line 32, different Z in line 124, and etc.
  • The text was reviewed and corrected. The manuscript has been reviewed and corrected by MDPI English Editing.
  1. the guiding descriptions in the template should be avoided, lines from 153-167.
  • The guiding descriptions were deleted.
  1. in the XRD patterns for BZn 8 and BZn10, one can observe many unassigned reflections around 35°, 42°, inconsistent to the assertion of phase-pure hopeite mentioned.
  • All peaks correspond to hopeite, but only the strongest peaks are highlighted. Now, the other peaks are identified, and the figure updated.
  1. for review in lines from 123-132, relevant Refs are needed.
  • Refs (16-18) have been added.

Reviewer 2 Report

This work by Abdulaziz et al. reports several minerals and performed many experiments to analyze their crystal structure. However, lacking a deeper analysis of these minerals leads to innovation is insufficiently, and relative results hard to support their conclusions. There are some comments to help you improve this manuscript.

1. The XRD curve of BZn2 shown in Figure 1 should be improved because the red line shows almost no peak can be recognized. The amorphous mineral would show a broad peak.

2. The language of this manuscript should be improved due to the large amount of non-academic descriptions.

3. The SEM images should be named Figure 2 rather than Figure 1.

4. The authors should give the method how to calculate the ratio between Ca and Zn in the mineral. However, the ICP experiment should be added, which could further confirm the ratio.

5. What information or conclusion do the authors want to draw from the XPS results?

The language of this manuscript should be improved due to the large amount of non-academic descriptions.

Author Response

We thank the reviewer for her/his careful reading of the manuscript and her/his constructive remarks. We have taken the comments on board to improve and clarify the manuscript. Please find below a detailed point-by-point response to all comments.

  1. The XRD curve of BZn2 shown in Figure 1 should be improved because the red line shows almost no peak can be recognized. The amorphous mineral would show a broad peak.
  • The XRD results were normalized, to obtain clear patterns, and the figure was updated. The JCPDS patterns of the major phases/minerals were added. (Please see Fig. 2)
  1. The language of this manuscript should be improved due to the large amount of non-academic descriptions.
  •      The text was reviewed and corrected. The manuscript has been reviewed and corrected by MDPI English editing service.
  1. The SEM images should be named Figure 2 rather than Figure 1.
  • The figures’ numbering has been reviewed and corrected throughout the manuscript.
  1. The authors should give the method how to calculate the ratio between Ca and Zn in the mineral. However, the ICP experiment should be added, which could further confirm the ratio.
  • A schematic diagram of the experimental design is added, please see Figure 1.
  • The ICP was carried out and we could detect only the Zn contents due to technical problem, (lease see Fig. 4) . Our aim at this stage of research is exploring and screening the new phases in general. We hope to carry out further research on optimizing the synthesis of these phases (i.e.: Hopeite and Parascholzite)
  1. What information or conclusion do the authors want to draw from the XPS results?
  • The conclusions of  the XPS are summarized. Please see lines 280-284 and lines 324-332

Reviewer 3 Report

Dear authors,

Your work is interesting, but some adjustments need to be made.

1. It is necessary to reduce the abstract to 200 words.

2. To increase readers' interest (Line 76), it should also be mentioned that bioactive coatings based on calcium and phosphorus are very common at the moment—for example, https://doi.org/10.1016/j.jmrt.2023.03.128.

3. For this work, it is also necessary to mention methods for studying the obtained crystals.

4. Clause 2.1 The manufacturer and country of production must be added to the used chemical reagents.

5. Line 153 – 166 This needs to be removed!!!

6. In clause 2.2, it is necessary to add a scheme of the chemical process for obtaining the given compositions.

7. Figure 1 - it is necessary to normalize the radiographs and present them in the proper form. The ICDD library numbers must be added to the obtained phases.

8. Line 237 Pattern repeats twice 1. Check the numbering of figures and tables.

9. It is necessary to expand the discussion using XRD data (Table 1) and the resulting morphology (Line 237). It is required to explain in more detail the effect of various additives on the formation of a particular phase and their morphological patterns.

10. XPS spectrum is incorrect—graph numbering from right (0 eV) to left (1200 eV).

11. P. 3.2 It is necessary to refer to some literary sources on the binding energies of the found elements.

12. Line 302 It is necessary to deconvolute the presented high-resolution spectra.

13. It is necessary to redo the conclusions. Add more comparison and specificity in the results.

Author Response

We thank the reviewer for her/his careful reading of the manuscript and her/his constructive remarks. We have taken the comments on board to improve and clarify the manuscript. Please find below a detailed point-by-point response to all comments.

  1. It is necessary to reduce the abstract to 200 words.
  • Done
  1. To increase readers' interest (Line 76), it should also be mentioned that bioactive coatings based on calcium and phosphorus are very common at the moment—for example, https://doi.org/10.1016/j.jmrt.2023.03.128.
  • A new paragraph about bioactive coating using CaPs is added. Please see [lines: 78-90], Refs: 16-18.
  1. For this work, it is also necessary to mention methods for studying the obtained crystals.
  • Methods as reported in 2.3, a schematic diagram of the experimental procedure is added; Figure 1.
  1. Clause 2.1 The manufacturer and country of production must be added to the used chemical reagents.
  • Done
  1. Line 153 – 166 This needs to be removed!!!
  • Done
  1. In clause 2.2, it is necessary to add a scheme of the chemical process for obtaining the given compositions.
  • Schematic diagram of the experimental procedure is added, please see Figure 1.
  1. Figure 1 - it is necessary to normalize the radiographs and present them in the proper form. The ICDD library numbers must be added to the obtained phases.
  • Figure 1 was updated accordingly. The peaks were normalized and the JCPDS of the of the main phases have been added.
  1. Line 237 Pattern repeats twice 1. Check the numbering of figures and tables.
  • The numberings of tables and Figures have been reviewed and corrected.
  1. It is necessary to expand the discussion using XRD data (Table 1) and the resulting morphology (Line 237). It is required to explain in more detail the effect of various additives on the formation of a particular phase and their morphological patterns.
  • Figure 4 with analysis is added to show the mineral transformation as a result of Ca replacement by Zn.
  1. XPS spectrum is incorrect—graph numbering from right (0 eV) to left (1200 eV).
  • Done, please see Fig. 5
  1. 3.2 It is necessary to refer to some literary sources on the binding energies of the found elements.
  • Done, references 40-42 were added.
  1. Line 302 It is necessary to deconvolute the presented high-resolution spectra.
  • Done, the figure was updated, please see Fig. 6
  1. It is necessary to redo the conclusions. Add more comparison and specificity in the results.
  • The conclusion section was reviewed and updated accordingly.

Round 2

Reviewer 1 Report

accept

Reviewer 2 Report

Agree to accept.

Reviewer 3 Report

Good job!